# Interhemispheric Pediatric Meningioma, *YAP1* Fusion-Positive

**DOI:** 10.3390/diagnostics12102367

**Published:** 2022-09-29

**Authors:** Silvia Esposito, Gianluca Marucci, Manila Antonelli, Evelina Miele, Piergiorgio Modena, Marzia Giagnacovo, Maura Massimino, Veronica Biassoni, Matilde Taddei, Marco Paolo Schiariti, Fabio Martino Doniselli, Marco Moscatelli, Luisa Chiapparini, Bianca Pollo

**Affiliations:** 1Pediatric Neuroscience Department, Fondazione IRCCS Istituto Neurologico Carlo Besta, 20133 Milan, Italy; 2Neuropathology Unit, Fondazione IRCCS Istituto Neurologico Carlo Besta, 20133 Milan, Italy; 3Department of Radiological, Oncological and Anatomo-Pathological Sciences, Sapienza University, 00185 Rome, Italy; 4Department of Pediatric Onco-Hematology and Cell and Gene Therapy, Bambino Gesù Children’s Hospital, IRCCS, 00165 Rome, Italy; 5Genetics Unit, ASST Lariana General Hospital, 22100 Como, Italy; 6Pediatric Oncology Unit, Fondazione IRCCS Istituto Nazionale Tumori, 20133 Milan, Italy; 7Department of Neurosurgery, Fondazione IRCCS Istituto Neurologico Carlo Besta, 20133 Milan, Italy; 8Department of Neuroradiology, Fondazione IRCCS Istituto Neurologico Carlo Besta, 20133 Milan, Italy; 9Department of Diagnostic Radiology and Neuroradiology, Fondazione IRCCS Policlinico San Matteo, 27100 Pavia, Italy

**Keywords:** *YAP1*-fusion, pediatric meningioma, interhemispheric meningioma, methylation profile

## Abstract

Meningiomas are uncommon in children and usually arise in the context of tumor-predisposing syndromes. Recently, *YAP1*-fusions have been identified for the first time as potential *NF2*-independent oncogenic drivers in the development of meningiomas in pediatric patients. We report a case of a *YAP1*-fusion-positive atypical meningioma in a young child and compare it with the previous ones reported. Extending the clinico-pathological features of *YAP1*-fused meningiomas, we suggest additional clues for diagnosis and emphasize the urgent need for an integrated multilayered diagnostic approach, combining data from histological and molecular analyses, neuroradiology, and clinical findings.

## 1. Introduction

Meningiomas are very common primary central nervous system tumors in adulthood, accounting for approximately 35% of all primary intracranial neoplasms, but are relatively rare in pediatric populations, comprising 0.4–4.6% of all pediatric brain tumors [1].

Pediatric meningiomas show unique features compared to their adult counterparts, both histologically and clinically (considering gender and site distribution) [2,3]. These lesions often occur in the context of tumor-predisposing syndromes, such as neurofibromatosis type 2 (NF2), schwannomatosis and multiple endocrine neoplasia type 1, of which NF2 is the most common [4]. 

However, the mutational spectrum of childhood sporadic meningiomas remains elusive to date: integrated genomic analyses have revealed a clonal predominance of the somatic *NF2* gene, confirming its crucial role in meningioma tumorigenesis, and some recurrent gene fusions in biologically relevant genes not well described up until now [3]. 

Recently, *YAP1* fusions have been identified for the first time as a potential *NF2*-independent oncogenic driver in the development of meningiomas predominantly in pediatric patients. To the best of our knowledge, only 10 *YAP1*-fusion pediatric meningiomas have been reported to date [5,6]. This report presents another case of a *YAP1*-fusion-positive atypical meningioma in a young child, suggesting additional clues for diagnosis and comparing it with the previous ones reported.

## 2. Case Report

A two-year-old child presented with two episodes of prolonged generalized seizures, expression of subcontinuous paramedian epileptiform activity. Brain magnetic resonance imaging (MRI) revealed a round solid mass with homogeneous enhancement located in the frontal interhemispheric space, tightly adherent to the anterior cerebral arteries, without evidence of leptomeningeal dissemination (approximate volume 9000 mm^3^, Figure 1b–d). No calcifications inside the tumor were detected on computed tomography (CT) images (Figure 1a). Minimal peripheral vasogenic edema was evident in frontal white matter. The mass was subtotally surgically removed, with minimal residual lesion attached to the walls of the anterior cerebral arteries. Postsurgical regularization of electroencephalographic signals and consequent seizure control were achieved. 

Microscopic examination revealed a cellular neoplasm, constituted of intersecting fascicles of atypical spindle cells, alternating with hypocellular fibrous nodules of dense collagenous tissue, and some areas characterized by discohesive rhabdoid-like elements (Figure 2a,b). The tumor cells showed mitotic activity (4 mitoses × 10 high power fields) and a proliferation label index ranging from 10% to 15%. The tumor cells invaded the brain tissue (Figure 2c). Immunohistochemical investigations did not provide a decisive immunoprofile, therefore methylation analysis of the tumor sample was performed. 

The methylation data of the tumor were categorized using the brain tumor classifier v11b4 and v12.5 (https://www.molecularneuropathology.org/mnp/, accessed on 7 June 2022) [7], which also generated a copy number variation (CNV) plot. The tumor clustered in the class meningioma with a 0.54 raw score (Appendix A) and optimal calibrated score (0.99), and as meningioma benign 3 (MNG_BEN_3) according to the v12.5 classifier (Appendix A). CNV analysis showed a possible gene rearrangement on chromosome 11q involving a region encoding for, among others, *MAML2* and *YAP1* (Figure 3) and a possible rearrangement on chromosome 22 not involving *NF2* gene (Figure 3 and Appendix A and Appendix B).

Therefore, a neoplastic fresh-frozen specimen was submitted to array-CGH investigation, revealing a complex chromosomal rearrangement of *YAP1-MAML2*, confirmed on RNA sequencing, suggestive for a chromothripsis event (Figure 4). Even with no familial history of signs and symptoms related to tumor-predisposing syndromes, genetic investigations (MLPA and NGS) were performed to rule out constitutional mutations in *NF2* and *SMARCB1* genes. The integrated diagnosis, combining biomolecular and histological findings, was that of *NF2* unrelated, sporadic, atypical meningioma (Appendix B).

At the first stabilized post-surgical MRI, three months after surgery, subtotal excision was confirmed (approximately volume 400 mm^3^, Figure 5a). At the one-year follow-up, radiological relapse was seen (approximately 700 mm^3^, Figure 5b), associated with electro-encephalographic worsening. Owing to residual progression which was unmanageable with surgery, the site of the disease, and the histological findings consistent with atypical meningioma (grade 2 WHO 2021 CNS5), proton beam irradiation was proposed.

## 3. Discussion

We report another case of a *YAP1*-fusion-positive atypical meningioma in a young child, after the original papers by Sievers [5] and Schieffers [6] in which *YAP1*-fusions were identified for the first time in 9/102 (8.8%) and 2/12 (16%) cases, respectively, as potential *NF2*-independent oncogenic drivers in the development of meningiomas, predominantly in pediatric patients (10/11 cases under 21 years of age). 

Immunophenotypic features of such cases are not straightforward: some of the previous reported cases were initially diagnosed as pleomorphic xanthoastrocytoma or pediatric high-grade glioma [5], and our case required methylation profile analysis. As shown in Figure 2a, our case showed cellular intersecting fascicles of spindle cells, alternating with hypocellular fibrous nodules of dense collagenous tissue, sharing these features with case #1 from the study by Schieffer et al. [6]. Furthermore, in our case some areas of rhabdoid cells were present (Figure 2b), a feature observed in two out of the nine *YAP1*-fusions in pediatric *NF2*-wildtype meningioma reported by Sievers et al. [5]. Such morphological clues, when observed in pediatric meningiomas, might be considered suspicious for a *YAP1*-fusion-positive pathway.

*NF2* tumor suppressor gene is an upstream negative regulator of the Salvador–Warts–Hippo pathway, an evolutionary conserved kinase cascade converging to *YAP/TAZ* transcriptional coactivators, and a fascinating pathway potently regulating several hallmarks in most solid cancers [8]. *YAP1* is a transcriptional co-activator and downstream effector of the Hippo pathway, and acts through TEAD transcription factors, regulating pro-survival and pro-proliferative transcriptional programs. *YAP1* activation could be promoted by loss-of-function mutations in core Hippo pathway upstream regulators, such as *NF2*, or mutations in genes encoding proteins that can impact the core Hippo signaling pathway. In contrast, while activating point mutations in the *YAP1* coding sequence are rare, recurrent *YAP1* fusion events have been identified in several subtypes of cancers [9]. 

YAP activation and subsequent deregulation of the Hippo pathway is a key mechanism in pediatric meningioma tumorigenesis; somatic *NF2* gene mutations remain predominant but recent advances identify *YAP1* fusions as potential NF2-independent alternative oncogenic drivers in these tumors [5].

*YAP1*-fusion meningiomas have been rarely reported, with 11 cases described to date, 10 of which were in patients under 21 years of age. The majority of these patients, including our case, harbored *YAP1-MAML2*, with the remaining four patients harboring *YAP1-PYGO1*, *YAP1-LMO1* and *YAP1-FAM118B*. No significant correlations with age, sex and WHO grade may be evidenced in this small patient cohort (Table 1). However, our case and 3 of the 10 patients reported in the previous series (30%) were atypical meningiomas, a quite high percentage, suggesting a negative prognostic inference of *YAP1* alterations which must be confirmed in a larger series.

In comparison with the other previously reported *YAP1*-fusion-positive atypical meningiomas, our case showed a particular midline location in the frontal interhemispheric space, adherent to the anterior cerebral arteries, hitherto never reported. Despite quite incomplete radiological findings in the described cases, lobar localization was the most common (4 out of 10 patients) (Table 1). Moreover, the location of the lesion in our case is quite atypical: the absence of evident dural attachment and the slight hyperdensity visible on CT, the cystic-necrotic appearance with marked and dishomogeneous enhancement and without a real dural adhesion noted on MRI suggested firstly an intra-axial malignant lesion. Meningiomas are usually attached to the dura mater deriving from progenitor cells that originate from the arachnoidal cap cells of the leptomeninges and only rarely and mainly in pediatric population have “intraparenchymal” meningiomas been reported: these tumors represent an unusual neuroradiological/neurosurgical finding, appearing similar to other more common intra-axial brain tumors [10,11]. The close relation of our case to the anterior cerebral arteries supports the hypothesis that intraparenchymal meningiomas arise from those arachnoid cells, located within the pia mater, which enter the surface of the brain and sulci, migrating with perforating blood vessels during brain development [11].

## 4. Conclusions

This case extends the clinico-pathological features of *YAP1*-fused meningiomas and highlights the need for an integrated multilayered diagnostic approach, combining data from histological and molecular analyses, neuroradiology, and clinical findings. Further studies in a larger series of pediatric *NF2*-unrelated meningiomas are needed in order to determinate the prognostic role of *YAP1* alterations and therapeutic advancements in this uncommon pediatric tumor.

## Figures and Tables

**Figure 1 diagnostics-12-02367-f001:**
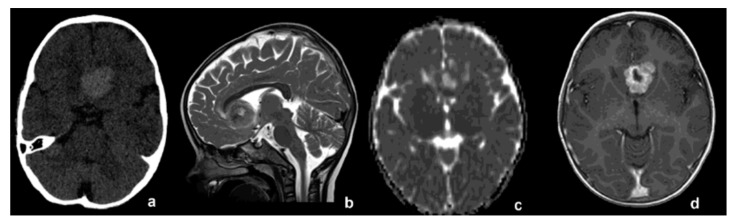
(**a**) CT without contrast medium, showing midline spontaneous hyperdense lesion. (**b**) T2-wi, (**c**) ADC map and (**d**) T1-wi after contrast medium. Pre-operative MRI showing close adhesion to anterior cerebral arteries.

**Figure 2 diagnostics-12-02367-f002:**
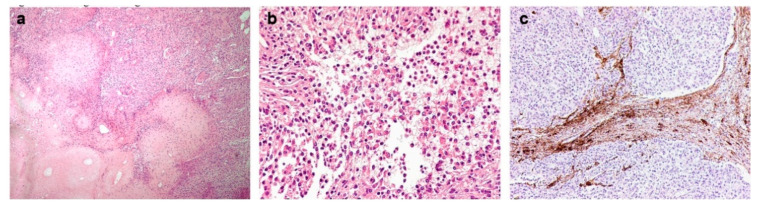
(**a**) The tumor was composed of cellular intersecting fascicles of spindle cells, alternating with hypo-cellular fibrous nodules of dense collagenous tissue (H&E, magnification 100×). (**b**) Some areas of rhabdoid cells were present (H&E, magnification 200×). (**c**) Immunohistochemistry for GFAP evidencing brain invasion (magnification 100×).

**Figure 3 diagnostics-12-02367-f003:**
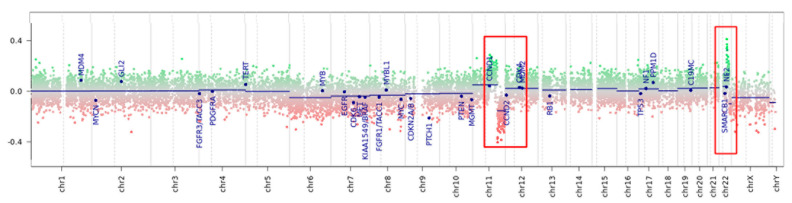
Copy number variation plot calculated from DNA methylation array data of the tumor sample. Depiction of structural rearrangements involving autosomes and X/Y chromosomes. Gains/amplifications represent positive (green), losses represent negative (red) deviations from the baseline. Twenty-nine tumor-relevant genomic regions are highlighted. Red boxes point out the rearranged region on chromosome 11q and on chromosome 22q (detailed in Appendix A).

**Figure 4 diagnostics-12-02367-f004:**
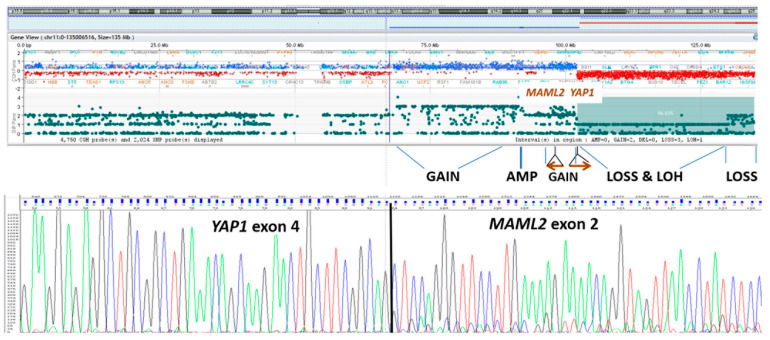
Array-CGH profile of chromosome 11, showing evidence of chromotrypsis with breakpoints located at *YAP1* and MAML2 gene loci. Sanger sequencing electropherogram demonstrating the presence of *YAP1*/MAML2 fusion transcript from tumor RNA.

**Figure 5 diagnostics-12-02367-f005:**
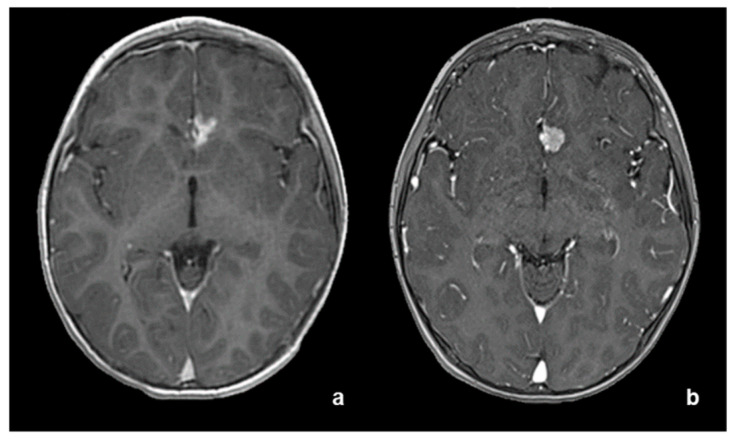
(**a**) Post-operative MRI at three months revealing minimal residual tumor and subsequent relapse at one-year follow-up (**b**).

**Table 1 diagnostics-12-02367-t001:** Clinico-pathological features of reported pediatric patients with YAP1-fusion-positive meningioma.

Case	Age (Years)	Sex	Tumor Location	MNG WHO Grade	Genetic Alteration
Present case					
1	2	M	Frontal interhemispheric space	II	YAP1-MAML2
Sievers et al. [5]					
2	4	F	Lateral ventricles, third ventricle	II	YAP1:MAML2
3	3	M	Temporal	I	YAP1:PYGO1
4	1	M	Third ventricle, lateral ventricle	NA	YAP1:MAML2
5	2	M	Skull base	NA	YAP1:MAML2
6	8	F	Skull base	I	YAP1:LMO1
7	17	M	Cavernous sinus	I	YAP1:MAML2
8	7	F	Parietal	NA	YAP1:MAML2
9	7	F	Frontal	I	YAP1:MAML2
Schieffer et al. [6]					
10	0, 57	F	Temporal	II	YAP1:FAM118B
11	20	M	Rolandic sulcus	II	YAP1:FAM118B

F: female; M: male; NA: no sufficient material for additional histological workup.

## Data Availability

The data presented in this study are available on request from the corresponding author.

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
