# Peer review of "Interhemispheric Pediatric Meningioma, YAP1 Fusion-Positive"

_diagnostics, 2022, doi:10.3390/diagnostics12102367_

Round 1

Reviewer 1 Report

It is a very interesting and attractive case. However, several points can be very important to document

1.- What is the frequency of genetic alterations (positive YAP1 fusion) in pediatric meningiomas.

two.-. Why was only YAP1- fusion performed in the current case?

2.- You mentioned earlier, the genetic alteration is not related to age, sex, histological grade, or the type of genetic alteration (due to the variety of fusions that YAP-1 expresses). Based on the other authors' findings, they did not report the ultimate implications for prognosis and/or treatment. It would be important if the work could give us information on the outcome of meningioma cases with this genetic alteration.

Author Response

Dear Reviewer, 

Hereafter is a list of point-to-point responses to yuor comments:

  • What is the frequency of genetic alterations (positive YAP1 fusion) in pediatric meningiomas?

Only recently YAP1 fusions have been identified as a potential NF2-independent oncogenic driver in the development of meningiomas, predominantly in pediatric patients.  To the best of our knowledge YAP1-fusion meningiomas have been rarely reported, with 11 cases described to date, 10 of which in patients under 21 years of age. In the original papers by Sievers and Schieffers, YAP1 fusion was found in 9/102 (8.8%) and 2/12 (16%) meningiomas respectively.  Welcoming the reviewer’s suggestion, we added in the text these percentages, underlying that they must be confirmed on larger series.

  • Why was only YAP1-fusion performed in the current case?

Thanks for this comment; however not only YAP1-funsion analysis was performed. This alteration was detected after a complex and extensive genetic work-up including DNA methylation profiling, CNV analysis and array-CGH investigation on neoplastic fresh-frozen specimen. Furthermore, genetic investigations (MLPA and NGS) were performed to rule out constitutional mutations in NF2 and SMARCB1 gene.

  • You mentioned earlier, the genetic alteration is not related to age, sex, histological grade, or the type of genetic alteration (due to the variety of fusions that YAP-1 expresses). Based on the other authors' findings, they did not report the ultimate implications for prognosis and/or treatment. It would be important if the work could give us information on the outcome of meningioma cases with this genetic alteration.

Thanks for the reviewer's comment. In the previous series reported by Sievers and Schieffers, there is no hint of patients follow-up. Concerning our patient, we observed a significant residual tumor progression unmanageable with surgery and proton beam irradiation was proposed.  This treatment option was chosen according to the progression, the site of the disease, and the histological findings consistent with atypical meningioma (grade 2 WHO 2021 CNS5), not according to the biological profile. To date, robust conclusions are not still derivable on the prognostic role, however we added in the text that 3 out of these 10 cases (30%) were atypical meningiomas, a quite high percentage which could signify a negative prognostic inference of YAP1 alterations which must be confirmed on larger series.

Reviewer 2 Report

The ms presents a clinical case report accompanied with mini-review of YAP1 fusion-positive interhemispheric pediatric meningioma. As a rare tumor, the case description provides one more example to the box, whereas the review summarizes up-to-date knowledge on the subject.

The analysis is well designed and meticously performed, especially the most important in that case - histopathological part. Very nice imaging documentation.

I would have just few remarks:

1. What are ELECTRO-clinical alterations? (57-58) That's rather uncommon expression.

2. 113 - one should rather not use colons - separate the sentence.

3. It would be of value to add short paragraph on YAP1 and fusion function and role in cancer, presence in ather tumors, connected with NF.

4. 156 - other studies are needed, but (because of tumor rarity) they are NOT URGENT. Such emotional adjectives should be avoided.

5. The paper is written in clear English - one thorough reading is needed to correct small punctuation, grammatical, and spelling mistakes.

Author Response

Dear Reviewer, 

Hereafter is a list of point-to-point responses to your comments:

  • What are ELECTRO-clinical alterations? (57-58) That's rather uncommon expression.

According to the reviewer's suggestion, we revised the sentences, underlining the positive effects of surgery on electroencephalographic signal and the consequent seizures control. At the same time, radiological relapse was associated with an electroencephalografic worsening.

  • 113 - one should rather not use colons - separate the sentence.

Thanks for the suggestion; we modified the sentence.

  • It would be of value to add short paragraph on YAP1 and fusion function and role in cancer, presence in other tumors, connected with NF.

We thank the reviewer for this suggestion. Thus we have clarified in the text the role and function of YAP1 and fusion in cancer and the emerging functional link between NF2, YAP1 and activation of the HIPPO pathway, whom deregulation seems to be a central mechanism in meningioma tumorigenesis

  • 156 - other studies are needed, but (because of tumor rarity) they are NOT URGENT. Such emotional adjectives should be avoided.

Thanks for the suggestion; we modified the sentence.

  • The paper is written in clear English - one thorough reading is needed to correct small punctuation, grammatical, and spelling mistakes.

Thanks for the reviewer’s suggestion. A comprehensive review by a native English speaker has been performed.